# Retrieving Vertical Cloud Radar Reflectivity from MODIS Cloud Products with CGAN: An Evaluation for Different Cloud Types and Latitudes

Fengxian Wang [1], Yubao Liu [1,*], Yongbo Zhou [1], Rongfu Sun [2], Jing Duan [3], Yang Li [1], Qiuji Ding [1] and Haoliang Wang [1]

[1] Precision Regional Earth Modeling and Information Center, Nanjing University of Information Science and Technology, Nanjing 210044, China

[2] Jibei Electric Power Corporation Limited, State Grid Corporation of China, Beijing 100031, China

[3] Key Laboratory for Cloud Physics of China Meteorological Administration, Beijing 100081, China

[*] Correspondence: ybliu@nuist.edu.cn; Tel.: +86-025-5823-5985

**Abstract:** Retrieving cloud vertical structures with satellite remote-sensing measurements is highly desirable and technically challenging. In this paper, the conditional adversarial neural network (CGAN) for retrieving the equivalent cloud radar reflectivity at 94 GHz of the Cloud Profile Radar (CPR) onboard CloudSat is extended and evaluated comprehensively for different cloud types and geographical regions. The CGAN-based retrieval model was extended with additional data samples and improved with a new normalization adjustment. The model was trained with the labeled datasets of the moderate-resolution imaging spectroradiometer (MODIS) cloud top pressure, cloud water path, cloud optical thickness, and effective particle radius data, and the CloudSat/CPR reflectivity from 2010 to 2017 over the global oceans. The test dataset, containing 24,427 cloud samples, was statistically analyzed to assess the performance of the model for eight cloud types and three latitude zones with multiple verification metrics. The results show that the CGAN model possesses good reliability for retrieving clouds with reflectivity > −25 dBZ. The model performed the best for deep convective systems, followed by nimbostratus, altostratus, and cumulus, but presented a very limited ability for stratus, cirrus, and altocumulus. The model performs better in the low and middle latitudes than in the high latitudes. This work demonstrated that the CGAN model can be used to retrieve vertical structures of deep convective clouds and nimbostratus with great confidence in the mid- and lower latitude region, laying the ground for retrieving reliable 3D cloud structures of the deep convective systems including convective storms and hurricanes from MODIS cloud products and used for predicting these storms.

**Keywords:** cloud retrieval; cloud radar reflectivity; CloudSat; MODIS; CGAN

## 1. Introduction

Clouds are an important component of the Earth's atmospheric system, which have significantly impacts on the radiation balance, water cycle, weather, environment, and climate change of the Earth's system [1]. However, there is a lack of observation techniques for detecting large-scale three-dimensional (3D) cloud structures, limiting the ability for understanding and forecasting clouds and precipitation [2]. At present, the passive satellite payloads only capture the horizontal two-dimensional distribution of some overall cloud features such as cloud optical thickness, the effective radius of cloud particles, cloud masks, and cloud water paths [3] at a synoptic or global scale. The Cloud Profile Radar (CPR) onboard CloudSat is a spaceborne active payload that detects cloud vertical structures, but with very minimal coverage and low temporal resolution.

The cloud vertical structures can be obtained by soundings, aircrafts, and radars. Zhou et al. (2013) studied the vertical structure of clouds in the Loess Plateau region using

a laser radar [4]. There have also been many attempts in the past to obtain information on the 3D cloud structure or reflectivity. Marshak et al. (2006) studied a 3D cloud droplet distribution by numerical methods and analyzed the liquid water content, number concentration of cloud droplets, effective radius, and extinction coefficient [5]. A recent experiment by Hilburn et al. (2021) showed that convolutional neural networks are able to obtain planar high radar reflectivity values from radiation gradients and the presence of lightning [6]. Barker et al. (2011) and Ham et al. (2015) used the data from two sensors to construct a 3D cloud field with an algorithm based on an assumption that the data from nearby vertical columns of similar radiance can be used to construct the target 3D cloud field [7,8]. Zinner et al. (2008) conducted a feasibility study on manufacturing cloud observation instruments to obtain high-resolution cloud effective radius [9]. Although the detection and retrieval of the cloud vertical structure have achieved some preliminary results, it is still very difficult to obtain large-scale cloud vertical structures.

Polar-orbiting satellites equipped with a Cloud-Aerosol Lidar with Orthogonal Polarization (CALIOP) radar of The Cloud-Aerosol Lidar and Infrared Pathfinder Satellite Observation (CALIPSO) and a CPR radar of CloudSat can accurately detect the vertical structure of cloud radar reflectivity. However, their spatial coverage is very limited, and their temporal resolution is low [10,11]. The Precipitation Radar (PR) of the Tropical Rainfall Measuring Mission (TRMM) and the Dual-frequency Precipitation Radar (DPR) of the Global Precipitation Measurement (GPM) can achieve single-frequency and dual-frequency rainfall particle detection and can thus obtain the vertical structure of precipitation. The horizontal resolution of the PR and DPR is 4.3 km and 5 km, respectively [12,13]. CloudSat is a polar-orbiting cloud observation satellite launched into space by the National Aeronautics and Space Administration (NASA) in 2006. Its millimeter-wave cloud measurement radar achieves vertical detection of clouds on a global scale [11]. Several scholars have analyzed the reliability of the cloud vertical radar reflectivity observed by CloudSat (Wang et al. (2010) [14], such as Matrosov et al. (2008) [15]). Welliver (2009) verified the cloud-bottom height data from CloudSat data using ground-based radar observations, and the results showed that the accuracy of CloudSat cloud-bottom height observations reached 73% [16].

Unlike the active payloads such as CloudSat/CPR, the passive payloads such as the Moderate-resolution Imaging Spectroradiometer (MODIS) have a capability to acquire large-scale and high-resolution 2D cloud information, including Cloud Effective Radius (Re), Cloud Optical Thickness (COT), Cloud Water Path (CWP), Cloudmask, Cloud top height (CTH), etc. [3]. In addition, geostationary satellites such as Himawari-8 and FengYun-4A/B monitor large-scale cloud systems with minute-level temporal resolution [16,17]. However, the passive remote sensing by visible and infrared radiances from these satellites cannot effectively penetrate optically thick clouds [18], making it difficult to observe cloud vertical structures. Several studies have attempted to obtain some vertical cloud features based on MODIS cloud parameters. For example, cloud analysis methods such as the Satellite Cloud and Radiation Property retrieval System (SatCORPS) divide clouds into three categories—high, mid, and low—and offer data on the total liquid and solid water content of each of these three groups, but these data still lack adequate and precise vertical cloud structure profiles (https://satcorps.larc.nasa.gov/, accessed on 13 December 2022). The acquisition of an accurate large-scale cloud vertical structure remains a pressing challenge for cloud-related research and applications.

The recent development of deep-learning technologies provides a feasible way to overcome these above problems. A GAN (generative adversarial network) model was proposed by Goodfellow et al. in 2014 [19], which uses two neural networks for adversarial training. GAN and its derivative models are commonly used to generate samples for data enhancement and data preprocessing. The generative model represented by the conditional adversarial neural network CGAN is suitable for the data retrieval purpose. CGAN is a branch of GAN, which can achieve the combination of input conditions and noise based on the input conditions to generate results. Leinonen et al. (2019) [20], hereafter L19, applied the CGAN model to retrieve the cloud vertical structure based on the MODIS data and

confirmed the feasibility. The model was trained using the matched CloudSat vertical radar reflectivity and MODIS cloud products including Re, COT, CWP, Cloud Top Pressure (CTP), and Cloud mask.

In order to promote the applications of L19's CGAN model for cloud analysis and data assimilation, a systematic assessment of the capability and reliability of the model is necessary. In this paper, based on the L19 model and the MODIS-L2 cloud products, we enlarged the dataset to train and test the model and perform a comprehensive validation study with a large volume of retrieval cases over the global oceans for the years 2010–2017.

We limit the cases to those occurring over the oceans, the same way as L19. A modification to the original model on data normalization is also introduced. The retrieval results of the test dataset were evaluated for different cloud types and latitudinal regions. The remaining part of this manuscript is organized as follows. Section 2 describes the data sources used in this study and the data characteristics. Section 3 describes the experimental framework, the CGAN retrieval model, the dataset, and the definition of the evaluation metrics. Section 4 gives the verification results. Finally, Section 5 summarizes the research results and discusses the limitations of the current technique.

## 2. Data

### 2.1. CloudSat Data

NASA CloudSat satellite products are available to the public through the CloudSat Data Processing Center website (www.cloudsat.cira.colostate.edu, accessed on 13 December 2022). CloudSat is in a sun-synchronous orbit at an altitude of 705 km, whose orbital period is about 99 min, about 14–15 orbits around the Earth per day. The CPR measurement resolution is 1.7 km along the track and 1.4 km across-track. Each vertical profile has 125 vertical layers with a resolution of 240 m, covering the ground to around 30 km altitude [10]. In this study, the radar reflectivity (at 94 GHz millimeter) and cloud classification information provided by CloudSat was used. The CPR radar differs from conventional weather radars in that it is 1000 times more sensitive than a standard weather radar that works at longer wavelengths. Unlike conventional weather radars, which are used to measure precipitation, CloudSat radars can detect small cloud particles. The CloudSat radar reflectivity maximum is only 20 dBZ.

This study uses CloudSat level 2 product 2B_GEOPROF and 2B-CLDCLASS data from 2011 to 2017 [10]. 2B_GEOPROF mainly provides vertical reflectivity (ref) information for clouds on the CloudSat orbit. CPR (3.2 mm) can penetrate relatively thick clouds and detect multiple cloud layers. However, when large particles exist, the detection capability of smaller particles may be reduced due to signal attenuation [10,11]. Furthermore, CPR has a good detection capability for optically thick clouds, but a weak capability for optically thin clouds. For example, Sassen and Wang [21] pointed out that CPR presents a difficulty in identifying thin cirrus clouds.

2B-CLDCLASS provides cloud classification information [22]. It classifies clouds into stratus (St), stratocumulus (Sc), cumulus (Cu, including cumulus congestus), nimbostratus (Ns), altocumulus (Ac), altostratus (As), deep convective (cumulonimbus), and high (cirrus and cirrostratus) clouds, based on different rules on the hydrometeor vertical and horizontal scales, the reflectivity measured by the CPR, indications of precipitation, and ancillary data including European Centre for Medium-Range Weather Forecasts (ECMWFs) predicted temperature profiles and surface topography height. It is worth pointing out that the cloud classification products of 2B-CLDCLASS have similar names and physical meanings as the corresponding conventional cloud classification but not surely consistent. Thus, we choose abbreviations or full names according to the official document that can be acquired at https://www.cloudsat.cira.colostate.edu/data-products/2b-cldclass#release-p1-r05 (last accessed on 5 January 2023; Tables 2 and 5).

### 2.2. MODIS Data

MODIS (https://modis.gsfc.nasa.gov/, accessed on 13 December 2022) is a space remote sensor developed by NASA, which is set on the Terra and Aqua satellites launched in 1999 and 2002, respectively. MODIS has 36 spectral bands covering the visible to thermal infrared bands, among which many atmospheric and cloud feature bands are included. A global observation (including daytime visible images and daytime and nighttime infrared images) is obtained every 1–2 days [12]. MODIS performs long-term global detection of atmospheric water vapor, cloud boundaries, cloud properties, and other information [18].

MODIS atmospheric level 2 standard products include cloud characteristic products: cloud volume, cloud mask, cloud phase, cloud microphysical parameters, cloud top parameters, etc. The products contain geographic coordinate information at a spatial resolution of 5 km, and cloud products such as cloudmask have resolutions of 1 km and 5 km. The cloud data above can be obtained from the MODIS product suites of MYD06 or MOD06 or MOD06-AUX [20]. MYD06 comes from the Aqua satellite, which is part of the same A-train as the CloudSat satellite (CloudSat traveled along the Aqua satellite orbit until 22 February 2018, approximately 460.286 km behind it with an average lag of 60 s behind). MOD06-AUX is the matching result of MYD06 and CloudSat by NASA.

A total of five MODIS L2 products are used in this study, namely cloud top pressure ($P_{top}$), cloud water path (CWP), cloud optical depth ($t_c$), effective radius ($r_e$), and cloud mask. $P_{top}$ characterizes the height of a cloud, CWP plays an important role in the radiative effect of clouds and the water circulation process in the ground–air system and can be divided into liquid water path (LWP) and ice water path (IWP) according to the phase of water. $t_c$ reflects the extent to which the cloud prevents the penetration of electromagnetic waves, and $r_e$ characterizes the scale of water particles within the cloud. The cloud optical thickness and the effective particle radius depend on the short-wave reflected radiation properties of the cloud. The cloud mask data used in this study are classified into four classes by calibration process: Cloudy, Uncertain, Probably Clear, and Confident Clear.

## 3. Experimental Design and Model Introduction

### 3.1. Experimental Design

Figure 1 is the working flowchart of this study. The steps for the cloud radar reflectivity retrieving process include (1) downloading the data from the official website, cutting the MODIS and CloudSat data into cells of 64 grid points in length, reducing the compressed or encoded data into real values, and matching them up. Next, we filtered out the nighttime and over-land data, and finally, we normalized the data of each element and divided the dataset into high, mid, and low latitudes (detailed in Section 3.2); (2) constructing the dataset (detailed in Section 3.4); (3) performing model training (detailed in Section 3.3) to obtain the trained network model; and (4) generating a large volume of inverse products and evaluate the model with multi-threshold multi-metric strategies (detailed in Section 3.5). In addition, based on the results in this paper, we developed a 3D-lization process to obtain 3D cloud reflection based on 2D MODIS data input and the result will be reported in a separate paper for the space limitation.

### 3.2. Data Pre-Processing

2B_GEOPROF and MOD06-AUX products for every tenth day from 2011 to 2017 were downloaded. The data are organized by grouping a full day of profiles as a single-day profile. A single 2B_GEOPROF profile and MOD06-AUX profile have 37,088 pixels with a resolution of approximately 1.1 km. The detailed data processing steps are as follows:

(1) Matching the MODIS data: MOD06-AUX data are processed the same way as Cloud-Sat data horizontally, both are intercepted into units of length 64 for this study. The 2B_GEOPROF data are divided into 64 slices vertically with a resolution of about 240 m. We selected 2B_GEOPROF with an altitude of about 700 m to 15,820 m for the model training. Each unit is referred to as a "scene" throughout this paper (following L19 [20]).

(2) Information filtering: Use the information within the cloudmask of MODIS data to filter by day/night: day, surface type: sea, cloud mask status: determined.

(3) Mask screening: The cloudmask is set to 1 when there is a cloud (Confident Cloudy by MODIS) and 0 for the rest of the cases, and the cloudmask of each pixel is also set to 0 when there is any MODIS variable with missing measurements. With this premise, if there are more than 32 cloudmask values of 0 in the 64 pixels of data, the data will be screened out. In order not to interfere with the training, we also screen out the cases where cloudmask marks clouds but there are almost no clouds.

(4) After steps (2) and (3), 162,125 samples were deleted and only 64,616 samples were retained. Then, we normalized the reflectivity data of 2B_GEOPROF data from [−27 dBZ, 20 dBZ] to [−1, 1], and the reflectivity smaller than −27 dBZ was set to −1 (because the reflectivity data of 2B_GEOPROF are disturbed by clutter and exhibit mosaic traits, which need to be filtered out). L19 suggested a normalization range of [−35 dBZ, 20 dBZ] in their public dataset, the normalized data are free of noise. We find that if normalization is performed with −35 dBZ as the minimum value, there are many noises, namely mosaic-like weak echoes in the cloud-free regions. Therefore, we adjusted the normalization using [−27 dBZ, 20 dBZ], which is effective to avoid noise interference. When normalizing the MODIS data, the missing data are substituted with a value of 0. The normalization for different variables is calculated by entering the following equation (Equations (2)−(5) are replicated from L19 for referring convenience):

$$Z'_{\mathrm{dB}} = 2\frac{Z_{\mathrm{dB}} + 27\ \mathrm{dB}}{47\ \mathrm{dB}} - 1 \tag{1}$$

$$P'_{\mathrm{top}} = \left(P_{\mathrm{top}} - 532\ \mathrm{hPa}\right)/265 \tag{2}$$

$$\tau'_{\mathrm{c}} = (\ln \tau_{\mathrm{c}} - 2.20)/1.13 \tag{3}$$

$$r'_e = (\ln(r_e/(1\ \mu\mathrm{m})) - 3.06)/0.542 \tag{4}$$

$$CWP' = \left(\ln\left(CWP/\left(1\ \mathrm{gm}^{-2}\right)\right) - 0.184\right)/1.11 \tag{5}$$

(5) After the dataset construction, we divided the global datasets by latitude into higher latitude (latitude > 65), mid-latitude (latitude between 20 and 65), and lower latitude regions (latitude < 20) for a comparison study of the model retrievals over different latitude regions.

*3.3. CGAN Convolutional Neural Network Model*

The deep learning network used in this study is a conditional adversarial network (CGAN) deep learning model by L19. The CGAN model is an extended model of GAN [23] which consists of two against parts, the Generator (abbreviated as *G* below), and the Discriminator (abbreviated as *D* below), both of which play roles dynamically in the training model to achieve the ideal result. The loss function of GAN is

$$\min_G \max_D V(D, G) = E_{\chi \sim P_{data}(x)} \log(D(x)) + E_{Z \sim P_z(z)} \log(1 - D(G(z))) \tag{6}$$

where $D(x)$ represents the probability of *D* estimating the real picture to be true, $D(G(x))$ represents the probability of *D* estimating the picture generated by *G* to be true, *E* represents the mathematical expectation, and when the data distribution $P_z$ is equal to the target distribution $P_{data}$, the best GAN can be obtained.

CGAN is an improvement on GAN, which adds conditional information to *G* and *D* of the GAN to achieve learning that uses the desired output with the corresponding conditional labels for training and generates specific products based on given conditional labels in the execution phase. The CGAN network training process in this paper is similar to L19, which is adapted in Figure 2 for convenience. In Generator *G*, the four inputs of the MODIS cloud elements, cloudmask data, and noise data are combined into a 64 × 64 data format and converted to a 8 × 8 × 256 data format by a full concatenation, a ReLU

activation function, and a batch normalization. After three 2D up-sampling processes and 2D convolution, a fake scene is generated. The discriminator $D$ input contains true and generated values of radar reflectivity. The input is converted to one-dimensional data by a fully connected layer after four layers of two-dimensional convolution and activation functions and finally, a judged data value is output after a sigmoid activation function. More details of the model can be found in the online source-code publication of L19 [20].

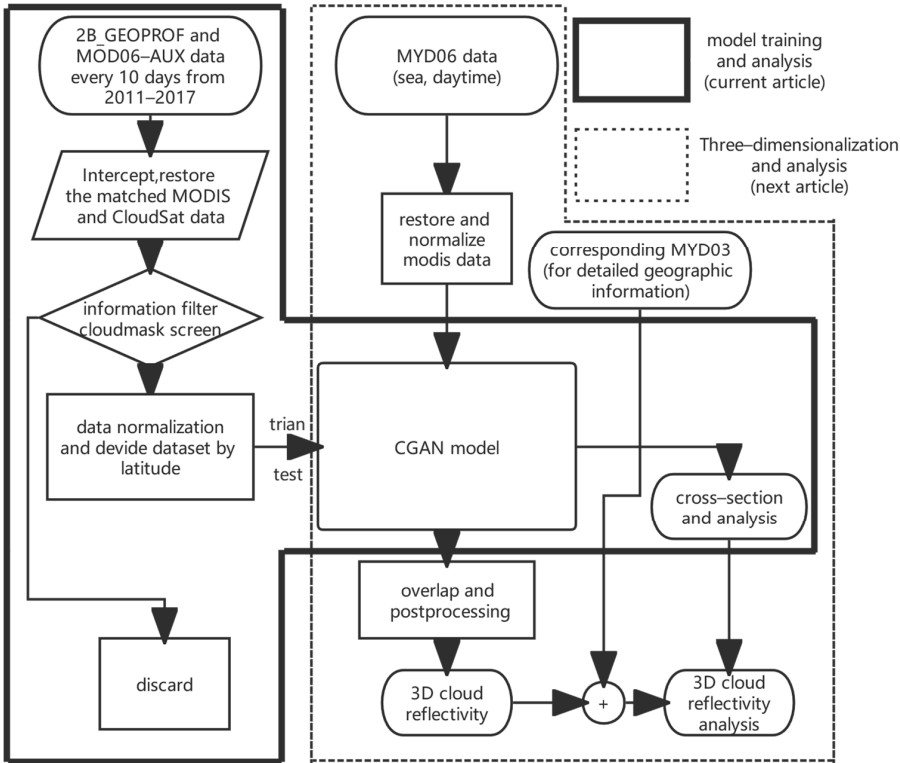

**Figure 1.** Implementation flowchart of the CGAN-based 3D cloud radar reflectivity retrieval technique. It is divided into a model training process and a 3D cloud map generation process. (MYD03 is a MODIS data file that provides detailed geolocation information, which is needed for comparison in the 3D cloud map generation step which is not discussed in this article. In this paper, we mainly discuss the model training process and the model evaluation.).

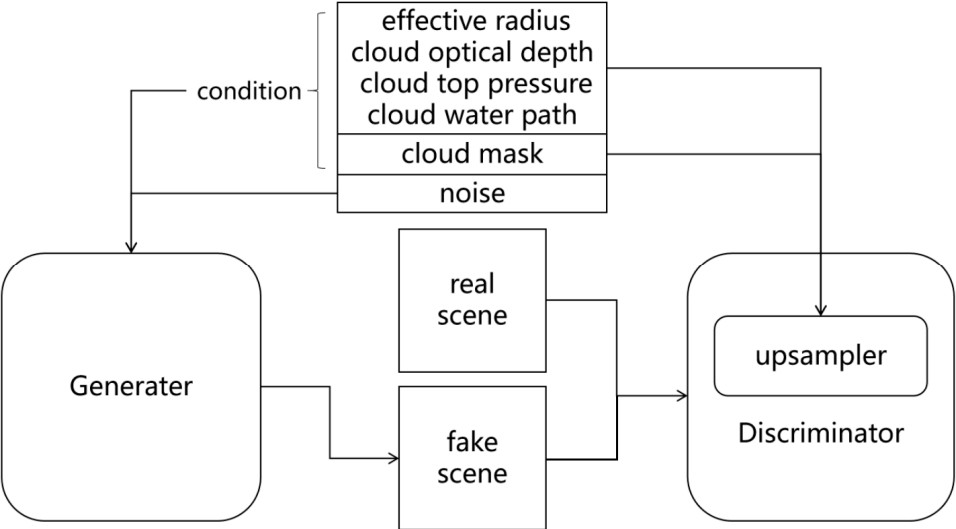

**Figure 2.** The framework of the Leinonen et al. [20] CGAN model for retrieving vertical radar reflectivity based on the MODIS cloud products.

*3.4. Training Set and Test Set*

The dataset of this study consists of a total of 179,660 scenes (labeled samples) over the global oceans for the whole year of 2010 from L19 [20] (hereafter referred to as old data) and 64,616 newly acquired scenes (hereafter referred to as new data), for a total of 179,660 + 64,616 items; 90% of them were selected as the training set, totaling 219,848 items; 10% of the new data were used as the test set, totaling 6461 items. Only new data are used for the test dataset because old data cannot offer cloud-type information. The new data are divided into the high, middle, and low latitude regions with 10,498, 13,634, and 40,494, respectively. A total of 10% of them were used as the latitudinal test set with 1049, 1363, and 4049, respectively.

*3.5. Evaluation Index of Retrieval Results*

In this study, five cloud radar reflectivity intensity thresholds and four statistical metrics are selected for a systematic evaluation of the CGAN model. The multi-threshold evaluation helps to evaluate different cloud radar reflectivity intensities, i.e., bulk cloud zones, cloud cores, etc., and multiple metrics assess different statistical aspects of the model. Given a selected threshold, the cloud area with radar reflectivity above the selected threshold is the true value (TRUE), and the model-generated cloud area exceeding the selected threshold is the predicted value (PREDICT). The overlap between the TRUE and PREDICT values is recorded as Hits, abbreviated as TP; predicted but not the truth is false-alarms, abbreviated as FP; not predicted but is the truth is misses, abbreviated as FN, and not predicted but also not the truth is correct-negatives, abbreviated as TN.

The verification metrics calculated in this study are:

(1) Threat score (TS) measures the fraction of observed and/or forecast events that were correctly predicted. It can be thought of as the accuracy when correct negatives have been removed from consideration, that is, TS is only concerned with forecasts that count. It is calculated as follows:

$$TS = \frac{TP}{TP + FP + FN} \tag{7}$$

(2) False alarm rate (FAR) indicates the proportion of actual cloud-free areas in the prediction to the total predicted cloud area, which is calculated as follows:

$$FAR = \frac{FP}{TP + FP} \tag{8}$$

(3) The probability of detection (POD) indicates the proportion of the predicted cloud occupying the TRUE cloud area, which is calculated as follows:

$$POD = \frac{TP}{TP + FN} \tag{9}$$

FAR and POD are often used in pairs and are used to complement each other to illustrate forecast capability.

(4) HSS (Heidke's skill score) indicates the accuracy of model forecasts after removing chance events, which can reflect the accuracy relative to the skill of random forecasts in a comprehensive manner. HSS is related to the threshold value, and a larger sample size is generally recommended. A larger HSS indicates better forecasts. A value of 0 indicates no skill, and HSS = 1 when the forecast is completely correct. Its calculation formula is

$$HSS = \frac{2 \times (TP \times TN - FN \times FP)}{\left(FN^2 + FP^2 + 2 \times TP \times TN + (FN + FP) \times (TP + TN)\right)} \tag{10}$$

L19 used the root mean squared error (RMSE) as an evaluation metric. Its calculation formula is as follows:

$$\text{RMSE} = \sqrt{\frac{1}{m}\sum_{i=1}^{m}(y_i - \hat{y_i})^2} \tag{11}$$

The reflectivity is expressed in decibel form and the RMSE is poorly representative of errors in weak echoes. Thus, unlike L19 which evaluated RMSE, we take TS, FAR, POD, and HSS as evaluation metrics.

## 4. Retrieval Results Testing and Evaluation

### 4.1. Case Analysis

For each of the eight cloud types, a typical retrieval example was selected and presented in Figure 3. All the cases shown are from the test set and they are selected by considering sampling in different global regions. These are relatively good cases included here for an illustration of the basic capability of the CGAN model.

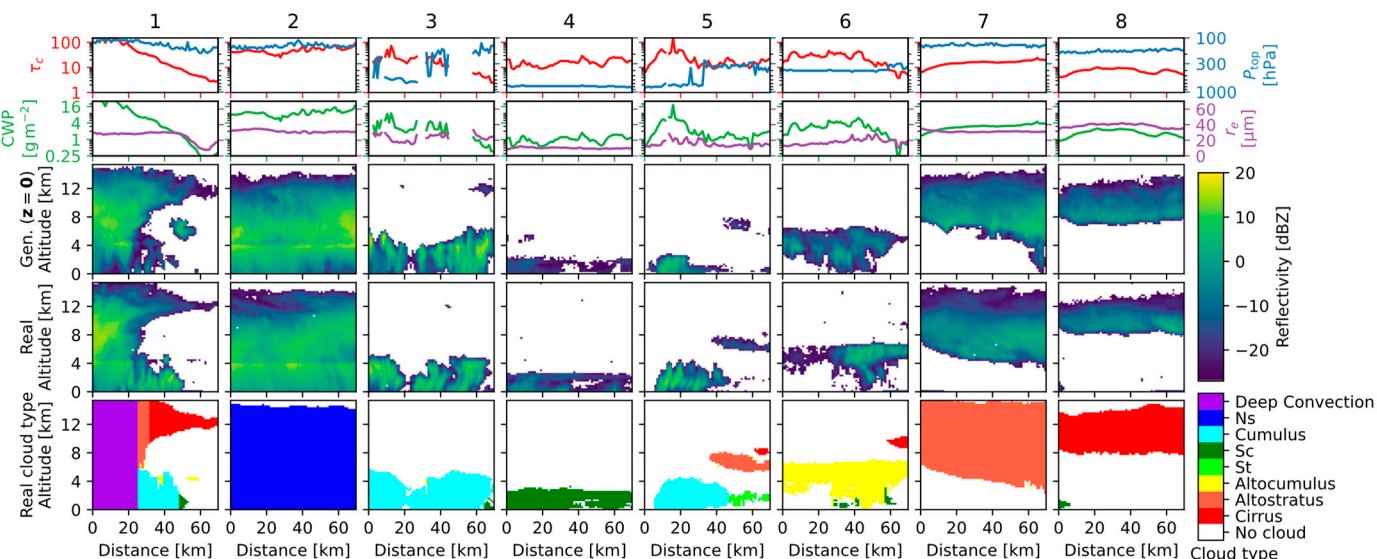

**Figure 3.** Examples of the cloud vertical radar reflectivity generated by CGAN for the eight main cloud types. Each column presents a typical case of one cloud type. The top two rows of subplots in each column are the cloud observations from MODIS (matching the CloudSat cloud radar observation tracks) including CWP, $t_c$, $r_e$, and $P_{top}$. The third row is the vertical radar reflectivity generated based on CGAN and MODIS data (noise input is 0), the fourth row corresponds to the actual CloudSat radar reflectivity, and the last row is the corresponding cloud classification labels. Some of the cloud body labels cover a slightly larger area than the displayed reflectivity image because the reflectivity image does not include reflectivity between [−35 dBZ, −27 dBZ] to avoid noise interference.

The first column of Figure 3 shows a deep convective cloud. Deep convective clouds are typically tall and have strong radar echoes. The deep convective cloud sample in Figure 3 has a typical anvil-like cloud top structure. There are dense cumulus clouds in the lower part of the periphery of the cloud body, and the CGAN model regenerated the basic characteristics of these clouds very well. The second column is the retrieval results for Ns, which has a large cloud range and fills the whole scene horizontally. The Ns cloud is also deep and has a strong reflectivity and the CGAN model exhibited a strong ability in retrieving this cloud. It retrieved the observed vertical structure of the cloud thickness and reflectivity intensity, including the structural characteristics of the bright band of the melting layer. The third column is for a cumulus, where the CGAN model not only generates the observed clouds but also simulates some details of the cloud texture. The fourth column is the retrieval results for Sc, where the CGAN model obtains the observed cloud body. The fifth column is the retrieval results for a low-level cloud (St). In general,

St clouds appear very rarely in the 2B-CLDCLASS products, and the clouds are generally small when they appear, so they are not well represented. The St clouds only appear with a little scale at the location of 50–64 km in the fifth column and are obscured by altostratus clouds and cirrus clouds, and thus, are badly inverted. Columns 6 and 7 are altocumulus and altostratus, respectively, and both belong to the mid-height cloud. The two samples in Figure 3 are properly retrieved by the CGAN model.

It is worth pointing out that in the 0–20 km region of the sixth column, the CWP and $t_c$ of the MODIS observations are pretty large, but the CloudSat observations are relatively weaker. Therefore, there may be data quality problems with this observation. Column 8 is a cirrus, and the thickness of the retrieval is a little larger, but the overall structure is similar to the observation and the height of the retrieval is very accurate. In general, although there is a small amount of noise and errors in the intensity and thickness of the cloud reflectivity in the CGAN retrieval, the overall cloud morphology is consistent with the CloudSat observations, demonstrating an overall promising skill of the CGAN model.

*4.2. Statistical Verification Results*

The dataset for the statistical evaluation consists of 6461 samples over the global oceans extracted every ten days from 2011 to 2017. The box plots of the statistic verification results for five thresholds and four metrics are given in Figure 4. Considering that using single samples for computing the statistics renders a too large number of samples and thus, results in a box plot with too large a span and reduced information characterization presentation, 16 samples are randomly selected to make a batch to calculate the average value, and the statistics are calculated with the batch as a unit.

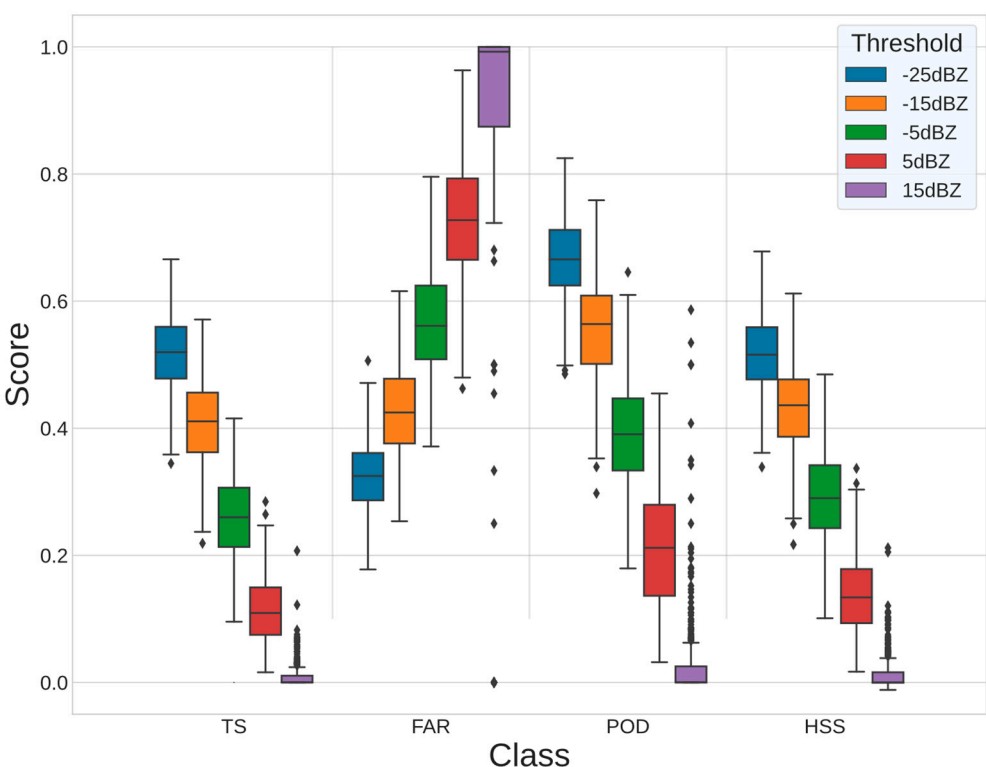

**Figure 4.** Multi-threshold multi-metric box plot of the test set. The horizontal axis is the four-evaluation metrics and the vertical axis is the score of each metric. The scores were calculated with five thresholds, shown in five colors. The outlier points are marked with diamond dots.

As can be seen in Figure 4, for the −25 dBZ threshold (which essentially characterizes the overall cloud body), the mean TS score is greater than 0.5, with an overall performance of 0.35–0.67, indicating that more than half of the overall extent of the "cloud" was predicted. The mean FAR value is around 0.3 and the mean POD score is around 0.65. Thus, the

overall predicted cloud area size is close to the actual one, but with a deviation of around 30% in position. The HSS mean value is greater than 0.5, indicating that after eliminating the correct predictions due to purely random chance, more than half of the predictions are measured correctly.

For the $-5$ dBZ threshold (representing the cloud areas with moderate reflectivity and above), the mean TS score is around 0.25 and the overall performance is between 0.1 and 0.4, indicating that about a quarter of the stronger cloud cores can be predicted. The mean FAR is around 0.55 and the mean POD is close to 0.4. Thus, the overall predicted cloud core area size is close to (slightly smaller than) the actual one, but there is a position displacement of about 55–60%. The mean HSS value is around 0.25. For the 15 dBZ threshold, representing an intensive cloud core region, the TS score, FAR, and POD are all relatively low and indicate poor retrieving ability. In general, the model exhibits an encouraging ability for retrieving the vertical structure of the clouds and presents promising application values.

There are several factors affecting the CGAN model's ability to capture the intense radar reflectivity cores: (1) The strong echo itself occupies a small area and low frequency of occurrence, resulting in insufficient training samples for the CGAN model; (2) CloudSat observation is affected by signal attenuation, especially when precipitating particles (e.g., rain drops, snow, graupel, and hail) were presented, and thus, its consistency with the MODIS cloud products is poor; and (3) The current model does not distinguish strong and weak echoes in the loss function during training, making the contribution of the strong reflectivity part to the loss function relatively small.

### 4.3. Statistical Evaluation for Different Cloud Types

Figure 5 shows the scatterplots of $-15$ dBZ and 0 dBZ sample POD vs. the sample size for all samples in the test set, overlapped with the sample-density heat maps. The data were divided according to the eight cloud types: deep convection, nimbostratus, cumulus, altostratus, stratocumulus, cirrus, altocumulus, and stratus. The cloud size is defined as the number of pixels it occupied. The pixel number is counted as the total number of pixels occupied by one cloud type in a single $64 \times 64$ pixel-sized scene. Due to the fact that when one type of cloud appears with a small size it may cause statistical interference, the figure removes the cloud examples with pixels less than 30. The metric POD is calculated by comparing the model-generated and the CloudSat-observed clouds by pixels. Each point in the scatterplot represents one cloud retrieval (scene). Evidently, the CGAN model presents a strong ability to retrieve the bulk cloud regions ($-25$ dBZ) and main cores (0 dBZ) of deep convective clouds. It also exhibits a good ability to retrieve the bulk cloud regions of nimbostratus and most altostratus and cumulus, but with larger errors for their main cores. The model has limited ability in retrieving cirrus, stratocumulus, and altocumulus, and behaves worst for stratus, partly because CloudSat does not detect stratus properly.

Figure 6 is the same as Figure 5 but summarized the distribution of the PODs for the test set samples for the eight cloud types. Again, the CGAN model exhibits an impressive ability in retrieving deep convective clouds and Ns, with little missing (i.e., zero POD value). The mapping relationship between the MODIS and CloudSat observations depends on cloud properties and the CGAN model indeed reflects the physical correspondence between the two. The clouds with a good physical agreement between the MODIS and CloudSat observations, e.g., deep convective clouds and nimbostratus, were well represented by the model and thus were retrieved accurately (Figure 6(a1,a2,b1,b2)). However, the other cloud types, e.g., cumulus, cirrus, stratus, etc., either with complex internal structures, small cloud size, or weak intensity, do not have a great correspondence between their MODIS and CloudSat observation counterparts, and thus the model does not work well (Figure 6(c1–h1,c2–h2)), with much lower scores.

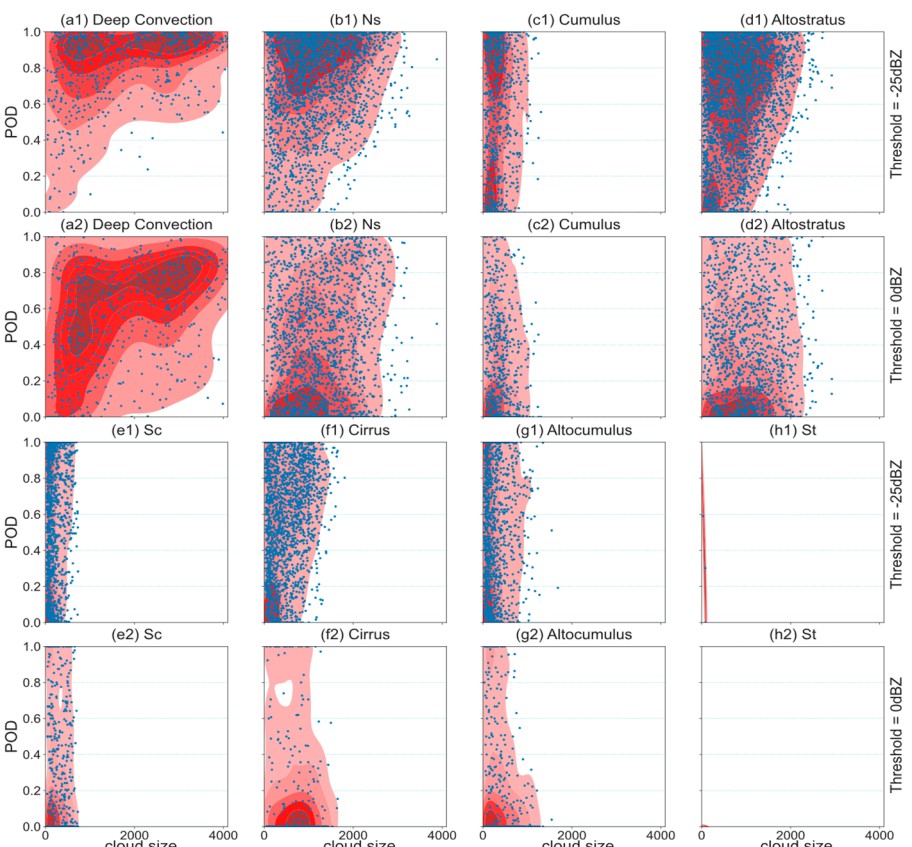

**Figure 5.** POD vs. size scatterplot of all-sample in the test set for thresholds of −25 dBZ (**a1**–**h1**) and 0 (**a2**–**h2**). The horizontal coordinate is the cloud size (i.e., the number of pixels covered by each inverse cloud sample region) and the vertical coordinate is the POD score. A point in each subplot indicates the POD result of a retrieved cloud sample. The color shades represent the sample distribution density heat maps.

As aforementioned, deep convective clouds are mainly cumulonimbus clouds, which are relatively large and thick. Deep convective clouds contain a significant amount of water from large droplets. The overall mismatch between the CloudSat measurements and the CGAN retrievals is minimal because these clouds satisfy all the favorable conditions for the retrieving model (Figure 6(a1,a2)). Ns clouds are slightly worse (Figure 6(b1,b2)). Altostratus, Sc, cirrus, and altocumulus are uniformly stratified but made up of small particles and cumulus clouds are small and thin. Thus, these types are a bit more difficult to retrieve than deep convective clouds and Ns and contain many missing elements (those with zero POD scores in Figure 6(c1–h1,c2–h2)).

### 4.4. Comparison of Cloud Retrievals at Different Latitudes

The Earth's atmospheric cloud systems vary with latitude. To assess the CGAN model's ability for retrieving clouds in different latitude zones, we divide the test set into high latitude (latitude > 65), middle latitude (latitude between 20 and 65), and low latitude (latitude < 20), and calculate the POD of the CGAN-retrieved clouds separately for each zone. The results a summarized in Figure 7.

In general, the model performed the best at the lower latitudes. For the Ns clouds, the model performed similarly for the mid- and lower-latitude zones but degraded significantly in the high latitude. The model retrieved the deep convective clouds accurately in both mid and lower latitudes, with a slightly better result in the mid latitude (Figure 7c). The model does not retrieve small-size deep convection properly in the high latitudes but is good with the large ones there. Cirrus in middle and high latitudes and altocumulus in high latitudes are poorly retrieved (Figure 7e). Except for the deep convective clouds and Ns, all other

cloud categories have evidently missed some retrievals (Figure 7a,b,d,f,g). The Sc clouds at high latitudes with smaller sizes are better retrieved than those at the middle and low latitudes, but the result may not be representative due to the small sample size (Figure 7b). Again, the samples of St cloud observations are too small to be considered (Figure 7h).

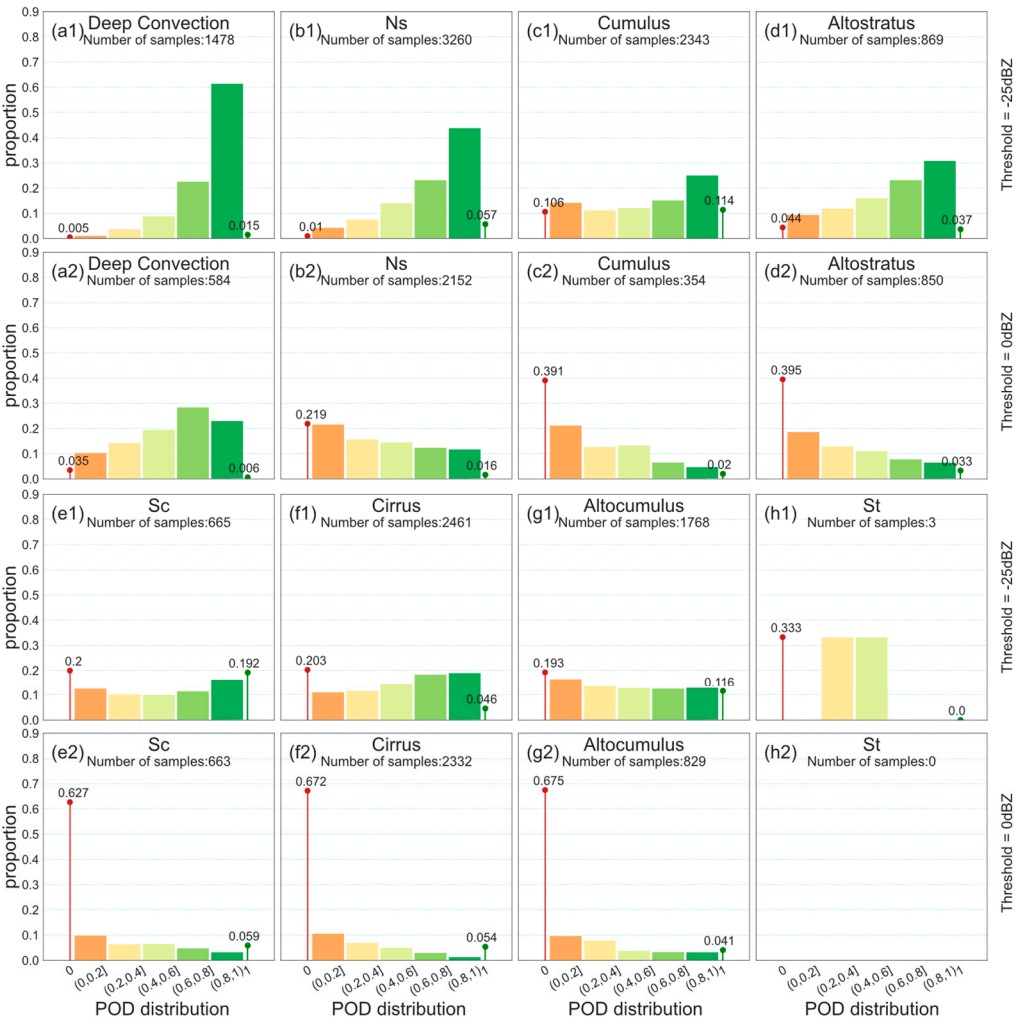

**Figure 6.** The distribution of POD scores for all test samples with threshold −25 dBZ (**a1–h1**) and 0 dBZ (**a2–h2**) for the 8 cloud types. POD values of 0 and 1 are plotted separately to indicate the two special cases of complete missing and complete forecasting, respectively. The percentage of POD falling in each interval is the vertical coordinate. "Number of samples" represents the total number of samples used to plot the statistics.

Figure 8 compares the frequency spectrum distributions of cloud radar reflectivity of the observed and the retrieved for the three latitude zones. In general, the CGAN model retrievals reflect the characteristics of the clouds at these latitudes very well, but the figure also exposes several significant features and deficiencies. The high value at the bottom line of the observation is because the lowest height is about 700 m, which may be a pixel (240 m) lower than L19.

Firstly, the CloudSat observations display significant differences in the radar reflectivity frequency distributions and their variations regarding height for the three latitude zones. The overall cloud radar reflectivity is stronger and taller at the lower latitudes, corresponding to the dominant deep convection activities in the region. These clouds contain a high hydrometeor water content. Strong reflectivity mainly occurred in the lowest 4 km layer. The CGAN model captures these cloud features very well.

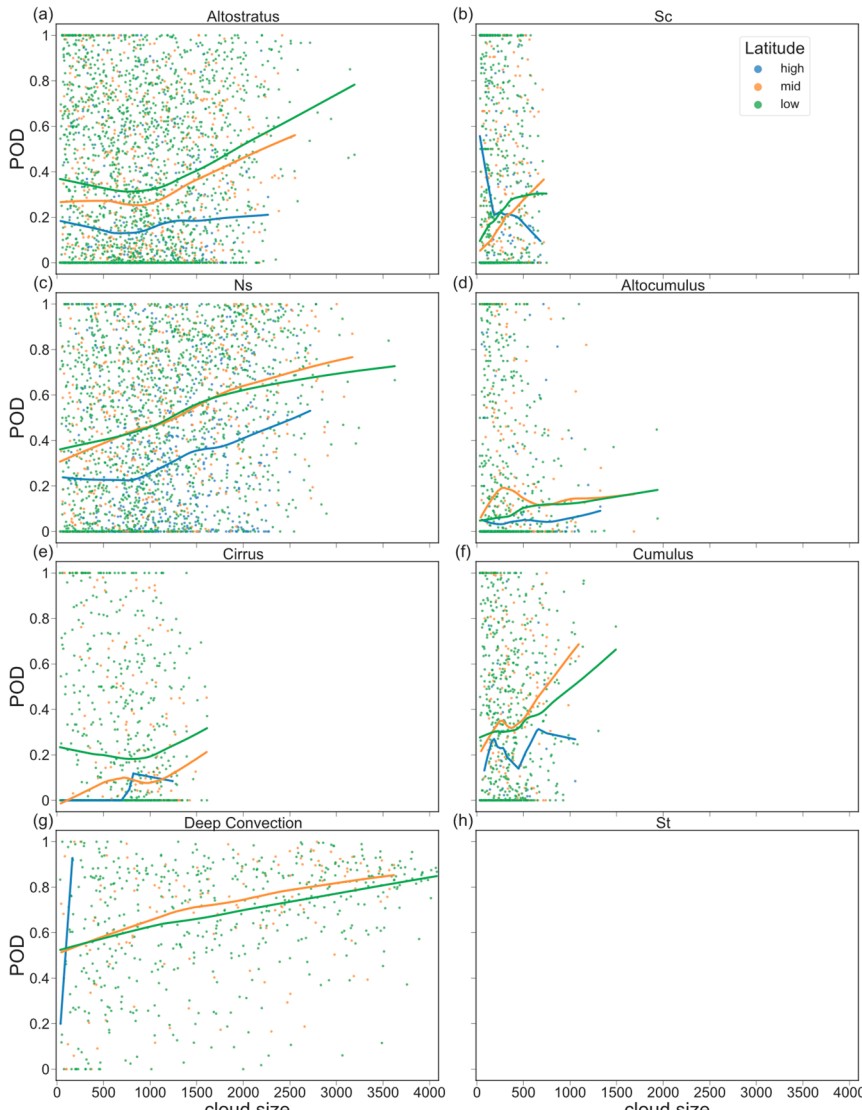

**Figure 7.** Same as Figure 5, but for the −5 dBZ threshold and the results of the lower, mid, and high latitude are distinguished by different colors. The curves corresponding to the colors are the least-square fitted lines.

Secondly, although the CGAN model retrieves the main features of the observed clouds above 4 km, it presents a systematic overestimation of both the radar reflectivity in the top region of the clouds and the cloud top height. This error increases significantly while moving to the higher latitudes, which is shown by the skewed positive radar reflectivity difference band in Figure 8i. The reason for this may be that the training dataset contains more samples from the lower latitude regions where the cloud is taller than those in the higher latitude regions.

Thirdly, the CGAN model underestimates the underlying strong cloud radar reflectivity. The error in the low and mid latitude regions is over 8 dBZ, and the maximum error is 10 dBZ, near the ground. The error extends upward to 1.5–2 km altitude in the lower latitude region and 4 km altitude in the mid latitude region. This problem is more serious at high latitudes, where the underestimation of strong reflectivity is more frequent throughout the 0–4 km layer. Finally, the CGAN model underestimates the occurrence of reflectivity less than −25 dBZ, which is particularly pronounced at high latitudes. In general, the CGAN model retrieves clouds at lower latitudes with smaller errors, degrades with increasing latitudes, significantly overestimates the occurrence frequencies of intense reflectivity clouds, and underestimates the weak ones.

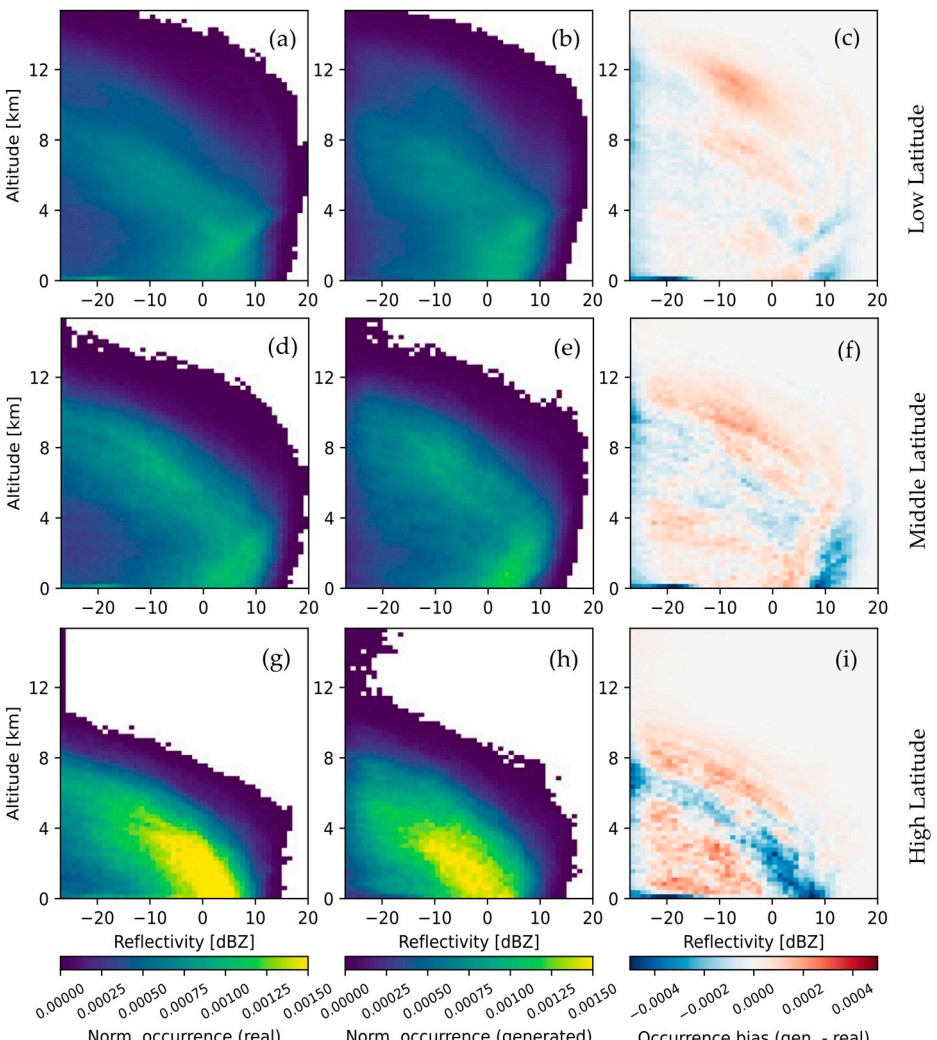

**Figure 8.** Normalized reflectivity–altitude distribution. The top (**a**–**c**), middle (**d**–**f**), and bottom (**g**–**i**) panels are for lower, mid, and high latitudes, respectively. The left (**a**,**d**,**g**), middle (**b**,**e**,**h**), and right (**c**,**f**,**i**) panels are the result of the real dataset, the generated dataset, and the difference between the two, respectively.

## 5. Conclusions and Discussion

In this paper, we extended the CGAN model developed by Leinonen et al. [20] for retrieving the vertical reflectivity profiles using the 2D MODIS cloud products. By increasing the training dataset, generating a new independent test dataset, and adjusting a normalization parameter of the CGAN model, we carried out a systematic evaluation of the performance and operational applicability of the CGAN model according to eight cloud types as classified by CloudSat and three latitude zones. The test dataset includes 24,427 samples distributed in different global oceanic regions during 2011–2017. The statistics were conducted with multiple verification metrics on multiple reflectivity thresholds spanning from the overall cloud bodies to high water content cloud cores. The conclusions are summarized below.

(1) The CGAN model can retrieve relatively realistic cloud vertical radar reflectivity structures, as well as cloud height and overall morphology, from MODIS satellite observations. The analysis of box plots with five cloud radar reflectivity thresholds (−25, −15, −5, 5, and 15 dBZ) and four metrics (TS, FAR, POD, and HSS) shows that the model possesses good skills for the low-reflectivity threshold representing the entire cloud bodies, but poor ones for recovering the high-reflectivity cloud cores.

This method can identify more than 50% of the cloud areas that are close to the observations, but there exist about 30% false alarms on average. The model can retrieve about 25% of the medium-intensity cloud cores of the observation accurately. For the strongest reflectivity cores, the model is only slightly better than the completely random prediction. There are three possible explanations for this result. Firstly, the strong-echo cores are small, with a low frequency of occurrence, i.e., too few samples to train the model properly. Secondly, CloudSat observations are affected by signal attenuation as well as the complexity of the intense clouds, and thirdly, the model is trained without distinguishing strong and weak echoes, making the contribution of the strong echo unintendedly less weighted. We will try to improve the representation of the loss function or apply a self-attention block to improve it in the future.

(2)  The CGAN model performance was evaluated for eight cloud types individually. The CGAN model performs the best for deep convective clouds, with more than 60% of the sample cases having POD scores greater than 0.8 at a threshold of $-25$ dBZ, and more than 50% of the cases having POD scores greater than 0.6 at a threshold of 0 dBZ. The model also exhibits a good capability for retrieving nimbostratus (Ns). About 50% of the samples have POD scores greater than 0.8 at a threshold of $-25$ dBZ, and more than 40% of the samples have POD scores greater than 0.4 at a threshold of 0 dBZ, just slightly worse than those for the deep convective clouds. The model also demonstrates some retrieval ability for altostratus and cumulus, with about 60% of the test samples having PODs greater than 0.5 at a threshold of $-25$ dBZ, and more than 60% of the samples having PODs greater than 0.5 at a threshold of 0 dBZ. The method is less effective for Sc, Cirrus, altocumulus, and St, mainly because MODIS and CloudSat have weaker or less consistent observations of these clouds. In general, the model is effective for clouds with regular structures, large thicknesses, and strong reflectivity (large particles).

(3)  The cloud systems at the lower latitudes are taller and denser than those at the middle and high latitudes, and the CGAN model works the best. The model performance decreases toward high latitudes, roughly worsening by 10% in the mid latitude, and a further 25% in the high latitude. The CGAN model performed the best for deep convective clouds for all latitudes and with good skill scores. For Ns, a similar effect is achieved by the CGAN model for the mid latitude and low latitude regions. The retrievals of cumulus and altocumulus clouds in the mid latitude perform slightly better than the low latitude regions. The CGAN model exhibits an overall good ability to recover the CloudSat observed occurrence frequency distribution according to reflectivity magnitudes and heights and their variations with the latitude. The model overestimates the reflectivity intensity in the upper regions and underestimates the intensity of the strong core at the lower layer, and these errors are the smallest at low latitudes and increase with latitude.

The performance variation of the CGAN model with cloud types and latitudes is related to the macro- and microphysical structures of the clouds, which determines the correspondence between the MODIS and CloudSat sensing of these clouds. Therefore, obtaining enough samples and using them to train the CGAN model for each cloud type, latitude zone, and/or even sub-region, individually, and independently or combined with the transfer-learning approach, would improve the model's retrieving skills. On the other hand, the structural design of the CGAN model proposed by Leinonen et al. [20] is set according to the computational resource requirements, and the parameter settings of this model, such as the sample level scale, may adversely affect the retrieval of the cloud regions close to the boundary. This can be improved by adjusting the model structure. The analysis also reveals that there are some serious inconsistencies between the MODIS CWP and the CloudSat reflectivity. Improving the data quality check or adding a physical constraint in the model would filter these problematic samples out and enhance the model's performance. We limit the cases to those occurring over the oceans the same way as L19, owing to the complexity of ground echo. Experiments over land can be studied further.

Finally, the model has a reflectivity overestimation of the upper cloud, for which we have no reasonable explanation at present. We are working on all the above problems and will report the findings in the future.

One of the most important findings of this work is that the CGAN model exhibits a great ability to retrieve the vertical reflectivity profiles of deep convective clouds and nimbostratus. These two types of clouds have the highest impact on human lives and activities because they are responsible for various severe weather conditions, including damaging winds, lighting, hails, heavy precipitation, and so on. Deep convective systems include convective thunderstorms, squall lines, mesoscale convective systems, and hurricanes/typhoons. We have extended the CGAN model for retrieving the 3D radar reflectivity structures for deep convective clouds and conducted more verification with ground-based weather radar observation. The 3D cloud structures of severe convective storms certainly present great potential for leveraging severe weather prediction capabilities through data assimilation of operational numerical weather prediction models and facilitating other convection forecast tools. This research will be conducted in future follow-up papers.

**Author Contributions:** Conceptualization, Y.L. (Yubao Liu) and R.S.; methodology, F.W. and Y.Z.; software, F.W. and Y.L. (Yang Li); validation, R.S. and Y.L. (Yubao Liu); formal analysis, F.W., H.W. and Y.Z.; investigation, F.W. and Y.L. (Yubao Liu); resources, Y.L. (Yubao Liu) and R.S.; data curation, J.D. and Q.D.; writing—original draft preparation, F.W. and Y.L. (Yubao Liu); writing—review and editing, Y.Z., Y.L. (Yubao Liu) and Q.D.; visualization, F.W.; supervision, Y.L. (Yubao Liu); project administration, Y.L. (Yubao Liu); funding acquisition, R.S. and Y.L. (Yubao Liu). All authors have read and agreed to the published version of the manuscript.

**Funding:** This research was supported by the Science and Technology Grant No. 520120210003, Jibei Electric Power Company of the State Grid Corporation of China and partially by the National Key R&D Program of China (2019YFC1510305).

**Data Availability Statement:** The original CloudSat data products 2B-GEOPROF and MOD06-AUX are available at the CloudSat Data Processing Center, http://www.cloudsat.cira.colostate.edu/ (accessed on 10 December 2022). A Python/Keras implementation code that can be used to reproduce the model is available at https://github.com/jleinonen/cloudsat-gan (accessed on 10 December 2022). The code to create the dataset and evaluate the model is available at https://github.com/SmartGragon/airef (accessed on 10 December 2022).

**Conflicts of Interest:** The authors declare no conflict of interest.

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
