# Peer review of "Retrieving Vertical Cloud Radar Reflectivity from MODIS Cloud Products with CGAN: An Evaluation for Different Cloud Types and Latitudes"

_remotesensing, doi:10.3390/rs15030816_

Round 1

Reviewer 1 Report

Review of Retrieving vertical cloud radar reflectivity from MODIS cloud products with CGAN: An evaluation for different cloud types and latitudes by Wang etc.

This paper studies the CGAN technique to retrieve cloud vertical structure from satellite observations. The sample size seems very limited. The paper does not state why only samples over globally oceanic regions are studied. The cloud vertical profile when it can be derived from the MODIS products will be more useful for storm predictions over land. Furthermore, ground-based radars over land will provide more training samples. The results show that the performance of the CGAN model varies with cloud types and latitudes, and the authors also mention that it may help when training the CGAN model individually base on cloud types and sub-regions independently. It will be better to emphasize why this kind of independent trainings is not done in the first place. Overall, this paper is well-written and I recommend to accept with minor revision.

Here is the list of minor comments/suggestions.

1.     Line 26: Paper shows the best result is achieved with reflectivity > -25 dBZ according to Figure 4. Why “reflectivity > 0 dBZ” is specifically mentioned here?

2.     Some abbreviations are not properly defined when they are first used in the main body (except for abstract). For example, “CPR” in Line 43, “CALIPSO” in Line 62, “TRMM”, “PR”, “GPM”, “DPR” etc. in Line 64, “MODIS” in Line 76, “SatCORPS” in Line 84, “MYD06”, “MOD06” and “MOD06-AUX” in Line 162.

3.     Line 77: Missing “,” after “(Re)”.

4.     Line 97: Could you check the reference [20] or [21]? “Leinonen et al. (2019)[20]”

5.     Line 137: The second stratus (Sc) may be stratocumulus”?

6.     Figure 1: “MYD03” appears suddenly. Is it a typo?

7.     Line 217: 162,125

8.     Line 243: Duplication “abbreviated as D”

9.     Line 461-465, Line 486, Line 489 and others: Could you keep use abbreviations or full name consistently, but not mix them. I prefer to use full names here. Anyway, the full names are not too long.

10.  Line 553-567: I trust the general results. Is there analysis (or Figure) supporting these conclusions? They do not appear in the main body. We usually do not introduce new information just in the conclusion section.

11.  Line 560-563: Could this sentence be more concise?

Author Response

Response letter to Reviewer #1

Overview:

This paper studies the CGAN technique to retrieve cloud vertical structure from satellite observations. The sample size seems very limited. The paper does not state why only samples over globally oceanic regions are studied. The cloud vertical profile when it can be derived from the MODIS products will be more useful for storm predictions over land. Furthermore, ground-based radars over land will provide more training samples. The results show that the performance of the CGAN model varies with cloud types and latitudes, and the authors also mention that it may help when training the CGAN model individually base on cloud types and sub-regions independently. It will be better to emphasize why this kind of independent trainings is not done in the first place. Overall, this paper is well-written and I recommend to accept with minor revision.

Answer:

We cordially thank you for your very constructive comments, helpful advice, and kind encouragement. All your comments and suggestions have been carefully considered in our revision of the manuscript and they are highly valuable for improving the quality of the paper. We believe our responses to your comments properly address your concerns.  

We fully agree with you that there is great potential to enhance and extend this work in several areas, including extending the work to inland regions. Nevertheless, in considering the good reliability of the MODIS cloud variables over oceans, we chose to start this work over oceans. We have started to test the 3D cloud retrieval model for the eastern China plain and got some good results too. We are also evaluating the model for remote regions in Northwest China, where weather radars are very sparse. It is very kind of you to suggest coupling MODIS data with ground-based radars and constructing model training according to cloud types and regions. These are what we are doing now. At present, the training dataset that we constructed is very limited and we are collecting more data to carry out the classified model training discussed above. We hope to report the results of these studies in a separate article soon.

Specific Comments:

  1. Line 26: Paper shows the best result is achieved with reflectivity > -25 dBZ according to Figure 4. Why “reflectivity > 0 dBZ” is specifically mentioned here?

Answer: 

Thank you for the comment. We attempted to use “reflectivity > -25 dBZ” to represent the whole cloud region and “reflectivity > 0 dBZ” to signify the cloud core regions. We made a mistake in the sentence where we intended to “reflectivity > -25 dBZ”, and it is corrected in the revised manuscript.

  1. Some abbreviations are not properly defined when they are first used in the main body (except for abstract). For example, “CPR” in Line 43, “CALIPSO” in Line 62, “TRMM”, “PR”, “GPM”, “DPR” etc. in Line 64, “MODIS” in Line 76, “SatCORPS” in Line 84, “MYD06”, “MOD06” and “MOD06-AUX” in Line 162.

Answer: 

Thank you for pointing out the issue. We have corrected them except “MYD06”, “MOD06” and “MOD06-AUX” in Line 162, because these product names are standard MODIS cloud product names, which are also described in Leinonen et al. (2019)[20]. In the revised manuscript, we added a reference pointer to the first mention of these names.

  1. Line 77: Missing “,” after “(Re)”.

Answer: 

Thanks. Corrected.

  1. Line 97: Could you check the reference [20] or [21]? “Leinonen et al. (2019)[20]”

Answer: 

Thank you. Corrected.

  1. Line 137: The second stratus (Sc) may be “stratocumulus”?

Answer: 

Thanks. It is stratocumulus. Corrected.

  1. Figure 1: “MYD03” appears suddenly. Is it a typo?

Answer: 

Thanks. MYD03 is a high-resolution geolocation field data file. We added a note about it in Figure 1’s caption.

  1. Line 217: 162,125

Answer: 

Thanks. Corrected.

  1. Line 243: Duplication “abbreviated as D”

Answer: 

Thanks. Corrected.

  1. Line 461-465, Line 486, Line 489 and others: Could you keep use abbreviations or full name consistently, but not mix them. I prefer to use full names here. Anyway, the full names are not too long.

Answer: 

Thank you for your suggestion. We apologize for not explaining this in the original manuscript. We choose abbreviations or full names to follow the official document “Level 2 Cloud Scenario Classification Product Process Description and Interface Control Document”, Table 2 and Table 5. The document is at:

https://www.cloudsat.cira.colostate.edu/data-products/2b-cldclass#release-p1-r05

which is referred. In the manuscript, we pointed out that the cloud classification products of 2B-CLDCLASS have similar names and physical meanings to conventional cloud classification but are not exactly consistent.

  1. Line 553-567: I trust the general results. Is there analysis (or Figure) supporting these conclusions? They do not appear in the main body. We usually do not introduce new information just in the conclusion section.

Answer: 

Thank you for your advice. Sorry for not being emphasized. The relevant discussion can be found at Line 394-411 in the revised manuscript.

  1. Line 560-563: Could this sentence be more concise?

Answer: 

Thank you for your advice. We modified this sentence from:

“thirdly, the model is trained without distinguishing between treating strong and weak echoes, making the contribution of the strong-echo part to the loss function relatively too small.”

to:

“thirdly, the model is trained without distinguishing strong and weak echoes, making the contribution of the strong echo unintendedly less weighted. We will try to improve the representation of the loss function or apply a self-attention block to improve it in the future.”

Response letter to Reviewer #2

Overview:

This study evaluates the conditional adversarial neural network (CGAN) in retrieving vertical cloud radar reflectivity from MODIS cloud products. The manuscript is well-written, with clear background introduction, method description, and result presentation. I have several comments on the manuscript.

Answer:

Thank you very much for your time, valuable comments, and kind encouragement. 

Specific Comments:

  1. The study uses optical thickness (OT), effective radius (r), and cloud water path (CWP) from MODIS cloud products. Indeed, CWP is almost proportional to the product of OT and r. Thus, CWP is not an independent variable from OT and r. What do the authors expect the CGAN performance if not using CWP in the training and testing process of CGAN?

 Answer: 

Thank you very much for your insightful opinion. We think CWP, COT, and Re represent different perspectives of cloud properties, especially for deep clouds. On the other hand, in AI/DL applications, by adding combined and/or derived meaningful variables in the input, one often improves the model performance. However, we agree with you that it will be interesting to conduct a “permutation importance” test to assess the contribution of the different input variables.

  1. In line 64, ‘TRMM’s PR’ and ‘GPM’s DPR’ do not have full spelling. The authors should also check other abbreviations and make sure the full spelling is given.

  Answer: 

 Thanks. Corrected.

  1. In line 173, to be clear, ‘waves’ should be ‘electromagnetic waves’.

 Answer: 

Thanks. Corrected.

  1. In line 200, why the datasets every tenth day are used rather than every day?

  Answer: 

 Thanks for your question. There is no special purpose for this. Because it is slow to download and process the data, we tried to cover more years for a limited time. We are now downloading more frequent data to extend our work in several other areas that will be reported in separate articles.

  1. In line 330, ‘Figure 11’ should be ‘Figure 3’?

 Answer: 

 Thank you very much. It is corrected.

Reviewer 2 Report

This study evaluates the conditional adversarial neural network (CGAN) in retrieving vertical cloud radar reflectivity from MODIS cloud products. The manuscript is well-written, with clear background introduction, method description, and result presentation. I have several comments on the manuscript.

1. The study uses optical thickness (OT), effective radius (r), and cloud water path (CWP) from MODIS cloud products. Indeed, CWP is almost proportional to the product of OT and r. Thus, CWP is not an independent variable from OT and r. What do the authors expect the CGAN performance if not using CWP in the training and testing process of CGAN?

2. In line 64, ‘TRMM’s PR’ and ‘GPM’s DPR’ do not have full spelling. The authors should also check other abbreviations and make sure the full spelling is given.

3. In line 173, to be clear, ‘waves’ should be ‘electromagnetic waves’.

4. In line 200, why the datasets every tenth day are used rather than every day?

5. In line 330, ‘Figure 11’ should be ‘Figure 3’?

Author Response

(The authors gave the same response as above.)
